# Sequence Requirements for miR-424-5p Regulating and Function in Cancers

**DOI:** 10.3390/ijms23074037

**Published:** 2022-04-06

**Authors:** Jiangying Xuan, Yingxia Liu, Xiaoping Zeng, Hongmei Wang

**Affiliations:** 1School of Basic Medical Sciences, Nanchang University, Nanchang 330006, China; xjy463399109@163.com (J.X.); welkin0120@foxmail.com (Y.L.); zeng-xp@163.com (X.Z.); 2Queen Mary School, Nanchang University, Nanchang 330006, China

**Keywords:** miR-424-5p, cancer, tumor microenvironment, tumorigenesis, chemotherapy

## Abstract

MiRNAs (microRNAs) are the most abundant family of small noncoding RNAs in mammalian cells. Increasing evidence shows that miRNAs are crucial regulators of individual development and cell homeostasis by controlling various biological processes. Therefore, miRNA dysfunction can lead to human diseases, especially in cancers with high morbidity and mortality worldwide. MiRNAs play different roles in these processes. In recent years, studies have found that miR-424-5p is closely related to the occurrence, development, prognosis and treatment of tumors. This review discusses how miR-424-5p plays a role in different kinds of cancers from different stages of tumors, including its roles in (i) promoting or inhibiting tumorigenesis, (ii) regulating tumor development in the tumor microenvironment and (iii) participating in cancer chemotherapy. This review provides a deep discussion of the latest findings on miR-424-5p and its importance in cancer, as well as a mechanistic analysis of the role of miR-424-5p in various tissues through target gene verification and pathway analysis.

## 1. Introduction

MicroRNAs (miRNAs) are a class of highly conserved endogenous noncoding RNAs in eukaryotic cells that contain approximately 20 to 25 nucleotides [1]. Since its discovery in 1993 [2], there has been a steady stream of functional studies on miRNAs. Currently, thousands of miRNAs have been identified and recorded in the online database miRbase (www.mirbase.org accessed on 13 March 2022). Based on studies of a variety of known human miRNAs, miRNAs are key regulators in various biological processes and play an important role in posttranscriptional gene regulation [3].

Mature miRNA is produced from long primary transcripts that undergo a series of cleavages by nucleases, followed by loading into the RNA-induced silencing complex (RISC). An miRNA recognizes target mRNA [4,5] through base complementary pairing. The degree of complementarity contributes to the silencing of the translation of the target mRNA by a complex degradation process or repression, which affects the expression of many genes through an interacting pathway. A single miRNA can target hundreds of mRNAs. MiRNAs regulate their targets through translation inhibition and mRNA instability. The 2–7 nucleotides at the 5′ end of miRNA are known as the miRNA seed region, which plays a vital role in the specificity of target binding [6]. MiRNA, which is considered a new gene regulatory factor, has been observed to participate in many human diseases, including neurodegenerative diseases, cardiovascular diseases [7] and diabetes [8]. In recent years, miRNAs have also been observed to play roles as oncogenes or tumor suppressor genes to participate in tumor cell proliferation, differentiation, apoptosis and other biological processes (Figure 1).

Cancer is actually the general name of more than 100 kinds of unique malignant tumors in various tissues of the human body. The abnormal level of miRNA in tumors has important pathogenic consequences, and is usually the result of the overexpression of miRNAs targeting tumor inhibitors or the downregulation of miRNAs targeting oncogenes [9]. MiRNAs directly or indirectly participate in the regulation of target genes and play an important role in key processes such as proliferation, migration and invasion in tumor cells. The abnormal expression of a variety of miRNAs can be involved in different tumors, while a single miRNA can act as an oncogene in one kind of cancer and as a tumor inhibitor in another. The dysregulated expression of miRNAs will cause most human tumors and malignant cells to show a dependence on miRNA expression [10]. Changing the expression of miRNAs will also change the expression of oncogenes and tumor suppressor genes, thus affecting the proliferation, apoptosis, motility and invasiveness of cancer cells, which provides an important opportunity for the development of new therapies based on miRNAs. MiRNAs have received more attention in the tumor treatment field as potential biomarkers for disease diagnosis, treatment and prognosis [11].

MiR-424-5p has been widely identified as a tumor suppressor gene that functions in many types of human cancer. It is processed from the 5′ end arm of the miR-424 precursor, is located on human chromosome Xq26.3 and is clustered with miR-15/miR-16. MiR-424 is a member of the miR-16 family. Because of its similar sequence structure and many common target genes and pathways, some researchers refer to it as the miR-16 family [5,12]. Some studies have found that this family can inhibit key genes to promote the transition between the G1 and S phases, regulating the cell cycle and cell phenotype. MiR-424-5p can target different genes in different tissues, indicating the diversity of its functions [13]. However, miR-424-5p is abnormally expressed in many kinds of malignant tumors [14,15,16,17], and its abnormal expression level is significantly related to the occurrence and development of malignant tumors. The expression level of miR-424-5p in tumors and its relationship with its corresponding target genes are summarized in this review. The mechanism of miR-424-5p in tumors will also be systematically analyzed, which can be concluded to involve roles in the regulation of gene expression, the tumor microenvironment and chemotherapy, suggesting that miR-424-5p may be used as a molecular biomarker in many tumors. At the same time, it is expected to provide new insight into molecular therapy for previously incurable tumors.

## 2. Regulation of miR-424-5p Expression in Cancer

MiR-424-5p is a member of the miR-16 family. The miR-16 family can induce G1 arrest by simultaneously regulating multiple downstream effectors [18]. CCND1 is a cell-cycle-related gene that plays a key role in the G1/S transition. In addition, it is a potential target for tumor gene therapy. A reverse screening method proved that CCND1 was regulated by the miR-16 family. Moreover, one study showed that the miR-16 family regulates the expression of several other cell cycle genes, including cyclin D3 (CCND3), cyclin E1 (CCNE1) and cyclin-dependent kinase 6 (CDK6). All these data suggest that the miR-16 family induces G1 phase arrest by simultaneously regulating multiple downstream effectors [19]. Additionally, miR-16 and miR-424 can regulate the expression of CCND1 by targeting putative target sites [18]. At the same time, the ectopic expression of miR-424 was shown to lead to a significant decrease in the numbers of key cell cycle regulators, such as cell division cycle 25A (CDC25A), cyclin A2 (CCNA2) and CCNE1. The miR-424 binding site was found in the CCNA2 3′-UTR, and miR-424 targets *CCNA2* directly by binding to the coding region of CCNA2, which can not only contribute to the differentiation of myoblast cells and cellular processes such as differentiation and epithelial-to-mesenchymal transition of cancer cells but also modulate the chemosensitivity of tumor cells toward anticancer drugs. Therefore, it can block the development of cancer [20]. Although various external stimuli affect miRNAs, the miR-16 family seems to be unique in its specific cell cycle-dependent regulation. The miR-16 family can regulate cell proliferation and/or apoptosis pathways in various tumor cells, participate in cell growth and inhibit cancer progression [19].

MiR-424-5p has different roles in different cancers and acts on different signaling pathways to regulate tumor progression (Table 1).

MiR-424-5p can be regulated by transcription factors (BFAR, bFGF, CCNE1J, MIEF2, E2F7) and regulatory proteins (WEE1, CARM1, DCLK1) in many cells. Moreover, the expression of miR-424-5p may also be affected by lncRNAs(long non-coding RNAs). In cancer cells, the abnormal regulation of these processes mentioned above leads to an imbalance in miR-424-5p, which leads to the occurrence and development of tumors.

There is also much evidence that miR-424-5p can be combined with lncRNAs. MiR-424-5p is predicted to have 354 binding sites for lncRNAs based on starBase data (http://starbase.sysu.edu.cn/ accessed on 13 March 2022). MiRNA and lncRNA are both noncoding RNAs. Generally, lncRNAs affect endogenous “miRNA sponges”, which inhibit the expression of miRNAs to influence the occurrence and development of tumors. Studies by Wang [42] and others have shown that by using 17 β-estradiol to stimulate LINC005511 expression, cell apoptosis can be inhibited. LncRNAs in the cytoplasm of ESR1-expressing ovarian cancer cells can compete with endogenous RNAs of miRNAs, including miR-424-5p, to interfere with the miRNA-mediated degradation of mRNA. This stimulates the proliferation of ovarian cancer cells. In non-small-cell lung cancer cell lines [29], lncRNA PVT1 can bind to miR-424-5p to decrease the expression of factors in tumor progression, such as CRAM, MMP-2, MMP-9 and Bcl2, to inhibit growth and increase the expression of Bax, thus suppressing the development of tumor cells. LncRNAs can also compete with other RNA transcripts as competitive endogenous RNAs (ceRNAs) to bind to the same miRNA to achieve regulation [43]. LncRNA FENDRR acts as a ceRNA [40], which directly targets miR-424-5p. In colon cancer, low FENDRR expression inhibits cell proliferation, migration and invasion by targeting miR-424-5p.

## 3. MiR-424-5p as a Tumor Suppressor

MiR-424-5p can inhibit tumor proliferation and migration by targeting certain genes (WEE1, E2F7, KIF23, MIEF2, CARM1, Notch) or regulating protein expression. The following cancers are discussed to demonstrate some related miR-424-5p mechanisms.

### 3.1. Hepatocellular Carcinoma

MiR-424-5p has been fully studied in hepatocellular carcinoma (HCC). Hepatocellular carcinoma is a malignant tumor derived from hepatocytes and hepatobiliary cells and is a common malignant tumor in China. With the increase in alcohol intake and the spread of hepatitis virus, China has become one of the countries with the highest incidence of liver cancer [44]. In these studies, it was found that the expression of miR-424-5p in tissues and cells of patients with HCC was decreased, and as verified by dual-luciferase reporter gene assays, the proliferation, invasion and migration of HCC cells were inhibited by targeting the inhibition of the expression of YAPI [45], TRIM29 [46], E2F7 [47] or ICAT [48]. Zhu [49] also found an abnormal increase in the level of the autophagy-related protein ATG14 in HCC tissues and cell lines. Bioinformatics results showed that X-inactivation specific transcript (XIST) could negatively regulate ATG14 by binding multiple miRNAs, including miR-424-5p, to reduce the proliferation and migration of HCC cells. Therefore, miR-424-5p can reduce the malignancy of hepatocellular carcinoma cells in HCC.

Some studies [50] have also found that downregulated DLX6-AS1, by targeting miR-424-5p, suppresses the expression of WEE1, which has been shown [21] to be a nuclear kinase that regulates the G2-M transition, and thus inhibition of WEE1 may be a potential targeted therapy for cancer by inhibiting the proliferation of HCC cells.

Some clinical studies also demonstrated [48] that compared with nonmetastatic HCC patients, the expression of miR-424-5p in liver cancer tissue and the serum of metastatic HCC patients was significantly lower and that the expression of miR-424-5p was lower in patients with a higher pathological grade and a higher TNM stage.

In recent years, an increasing number of studies have investigated the signaling pathways related to miR-424-5p in HCC cells. Meta-analysis by some scholars [51] showed that PVT1 reduced the expression of miR-424-5p through the regulation of the p53 signaling pathway, thus affecting carcinogenesis and promoting the proliferation of HCC. Some studies have also shown that [22] lncRNA MYLK-AS1 upregulates E2F7 by targeting miR-424-5p directly to activate the VEGFR-2 signaling pathway, thus promoting angiogenesis and cell proliferation of HCC tumors in vivo and in vitro.

As mentioned above, lncRNAs can regulate other RNA transcripts by targeting specific miRNAs. LncRNA CDKN2B anti-sense RNA 1 (CDKN2B-AS1), which is an anti-sense RNA of cyclin-dependent kinase inhibitor 2B (CDKN2B), plays an important role in many diseases, including cancer. It was demonstrated in [52] that the expression of miR-424-5p in HCC decreased, while lncRNA CDKN2B showed the opposite expression. A dual-luciferase assay verified that the target gene of lncRNA CDKN2B-AS1 was miR-424-5p. The overexpression of miR-424-5p decreased cell viability, inhibited the cell migration and invasion capacities, and affected epithelial-mesenchymal transformation (EMT). Some studies [53] have also shown that miR-424-5p can be identified as the downstream target gene of lncRNA CASC9. LncRNA CASC9 promotes the proliferation, migration, invasion and apoptosis of HCC cells in vitro by negatively regulating miR-424-5p.

Some experimental results showed that miR-424-5p could reverse the recovery of malignant behavior in HCC, which further confirmed the anticancer function of miR-424-5p in HCC (Figure 2). This provides a novel and promising treatment strategy against liver cancer.

Another type of liver cancer is intrahepatic cholangiocarcinoma (ICC) [54], which is a malignant tumor originating from intrahepatic bile duct epithelial cells, and its incidence has increased significantly in recent years. Only a few studies [23] have shown that the expression of miR-424-5p is downregulated in ICC and inhibits mTOR phosphorylation through targeted regulation of ARK5 in vitro, thus inhibiting the migration, invasion and epithelial-mesenchymal transition of ICC cells but having no effect on proliferation. This may be a new way to inhibit ICC metastasis.

In summary, miR-424-5p is expressed as a tumor suppressor in liver cancer, and the recovery of its expression may be a promising strategy for the treatment of liver cancer.

### 3.2. Ovarian Cancer

Ovarian cancer (OV) is one of the most common malignant tumors of the female reproductive organs. Its morbidity and mortality account for approximately 4% of all cancers in the world [55], second only to cervical cancer and uterine body cancer. Among these, epithelial ovarian cancer (EOC) has the highest mortality rate among all the types of gynecological tumors and poses a serious threat to the lives of women. OV has always been considered to be a hormone-dependent cancer, and therefore hormone receptors have become the primary research focus for targeted therapy for ovarian cancer [56]. Studies by Wang [42] and others have shown that the estrogen receptor (ESR1) transcriptionally regulates LINC00511 expression. Upregulated LINC00511 stimulates the proliferation of ovarian cancer cells by inhibiting apoptosis, while this long non-coding RNA inhibits the expression of a series of miRNAs including miR-424-5p and increases the expression of oncogenes. Therefore, miR-424-5p shows an active anticancer effect in OV. In addition, some studies have shown that miR-424-5p negatively regulates tumor-induced hypertrophy by directly targeting acyl-CoA synthetase long chain family member 4 (ACSL4) in OV cells, and subsequently reduced erastin- and RSL3-induced ferroptosis. This provides a new idea and method for OV therapy [24].

The metastasis of ovarian cancer is closely related to mortality, and therefore the biological characteristics of the tumors are also being actively explored. Liu et al. [57] showed that compared with normal tissues and cell lines, the level of miR-424-5p was significantly downregulated in EOC, which was negatively proportional to the TNM stage, tumor size and metastatic degree of EOC. The experimental results showed that miR-424-5p led to the arrest of the cell cycle in the G1 and S phases by targeting the CCNE1-mediated E2F1-pRb signaling pathway, which led to an antitumor effect on ovarian cancer. Similarly, Tong et al. [25] confirmed that the miR-424 cluster was significantly decreased in ovarian cancer due to hypermethylation of its promoter, in which miR-424-5p directly inhibited KIF23, while KIF23 promoted cell proliferation and migration in vitro. The knockdown of KIF23 affected the distribution of the cell cycle, the percentage of cells in G1 phase increased significantly, and the percentage of cells in S phase decreased significantly. It is suggested that the decrease in miR-424-5p can enhance the expression of KIF23 and inhibit the proliferation and migration of ovarian cancer cells. Recently, it has been shown [58] that the downregulation of miR-424-5p leads to the overexpression of MIEF2 (mitochondrial elongation factor 2) in OV tissues and cell lines, while the overexpression of MIEF2 significantly promotes the metabolic transition from oxidative phosphorylation to glycolysis in OV cells, and that the change in glucose metabolism characterized by increased glycolysis (also known as the Warburg effect) has been recognized as one of the markers of cancer [59] This leads to the occurrence, development and metastasis of tumors in OC.

Similarly, Cha et al. [60] showed that miR-424-5p is also a tumor suppressor in serous ovarian cancer. The expression of miR-424-5p in ovarian cancer was significantly lower than that in the normal group, and the decrease in miR-424-5p expression was significantly related to distant metastasis. Therefore, it is suggested that the decreased expression of miR-424-5p in ovarian cancer may be a recognized biomarker of distant metastasis of ovarian cancer.

From this series of studies, it can be suggested that miR-424-5p can target a variety of oncogene mRNAs and play a tumor inhibitory role in OV (Figure 3).

### 3.3. Cervical Cancer

Cervical cancer is one of the most aggressive cancers in women, with a high recurrence and mortality [61]. Previous studies have shown that the expression of miR-424 in cervical cancer tissue is decreased, which inhibits the proliferation and growth of cancer cells [62,63]. MiR-424-5p is a branch of the miR-424 family, and thus it has been studied in great detail. Dong [64] showed that lncRNA nuclear RNA host gene 12 (SNHG12) was abnormally elevated in human cervical cancer tissue, while silencing SNHG12 inhibited the proliferation of cervical cancer cells and tumor growth in a nude mouse model. By predicting that SNHG12 has a presumptive binding site related to the seed sequence of miR-424-5p, the experimental verification showed that the tumor-promoting effect of SNHG12 was through acting as a molecular sponge of miR-424-5p, thus negatively regulating the expression of miR-424-5p in cervical cancer to promote the proliferation and invasion of cervical cancer cells. Similarly, Zhou [26] demonstrated that the expression of miR-424-5p was significantly decreased in cervical cancer tissues and cells, while a dual-luciferase assay confirmed that KDM5B was the direct target gene of miR-424-5p. MiR-424-5p inhibited cell proliferation and migration and promoted apoptosis in cervical cancer cell lines by targeting the KDM5B-Notch pathway. The cervical squamous cell carcinoma mentioned above, which is a type of cervical cancer, demonstrated that miR-424-5p had anticancer effects. Therefore, miR-424-5p plays an overall role in inhibiting the occurrence and development of cervical cancer, which provides a new idea and scheme for clinical treatment in the future.

### 3.4. Neurological Malignancies

Glioma is one of the most common primary tumors of the nervous system, accounting for approximately 40–50% of intracranial tumors. Although great progress has been made in treatment in recent years, the prognosis of gliomas is poor due to the special properties of gliomas, such as invasive growth, recurrence, radiotherapy and chemotherapy resistance and other factors [65]. Therefore, gene targeting research provides a new path for treatment. It was concluded that bifunctional apoptosis regulator (BFAR) is a direct target of miR-424-5p. Cheng [27] showed that the content of miR-424-5p was decreased significantly in glioma tissues. As predicted by TargetScan and confirmed by bifunctional luciferase reporter gene analysis, BFAR is expressed negatively proportionally to miR-424-5p. MiR-424-5p could inhibit the progression of gliomas by decreasing Ki-67 expression, thus inhibiting invasion and proliferation.

Similarly, high levels of ALK protein in neuroblastoma (NB) are associated with metastatic NB cases and poor prognosis, while experimental results [66] have shown that miR-424-5p can directly target ALK receptors or indirectly regulate ALK expression in NB cells, resulting in a significant decrease in ALK protein and inhibition of cell viability in ALK-positive NB cell lines.

Zhou et al. [67] found a similar situation in pituitary neuroendocrine tumors (PitNETs). The specific transcriptional gene product of lncRNA X inactivation upregulates the expression of basic fibroblast growth factor (bFGF) by competitively binding to miR-424-5p to inhibit the proliferation, migration and invasion of invasive PitNET cells and promote cell cycle arrest and apoptosis of invasive PitNET cells.

Therefore, miR-424-5p may be a favorable supplement for combined therapy for the treatment of nerve cell malignant tumors.

### 3.5. Breast Cancer

Breast cancer (BC) is one of the most common cancers and therefore its pathogenesis and treatment have been the focus of much research. In recent years, there have been an increasing number of studies on targeted gene therapy. Experiments have proven that miR-424-5p is closely related to the occurrence and development of BC. Some experiments have shown that the overexpression of miR-424-5p in BC can inhibit cell proliferation and migration. A year later, Narges et al. [28] also confirmed the antitumor activity of miR-424-5p in BC cells and proposed the corresponding mechanism, which is that miR-424-5 can target PD-L1 and regulate the expression of the PTEN/PI3K/AKT/mTOR signaling pathway to inhibit cell proliferation and lock the cell cycle in the G2 phase. Another way that miR-424-5p inhibits tumor cell proliferation and arrests cells in G2/M cell phase is to activate the Hippo pathway and the extracellular signal-regulated kinase pathway by suppressing CDK1 mRNA in human breast cancer [68]. Similarly, brain-derived neurotrophic factor (BDNF) was proven to be the direct target protein of miR-424-5p in BC cells, and LINC00922 was shown to regulate the expression of BDNF by sponging miR-425-5p. LINC00922 is obviously overexpressed in breast cancer tissues and cell lines, and therefore increasing the expression of the miR-424-5p/BDNF axis can reduce the expression of LINC00922 and weaken its effects on promoting the proliferation, apoptosis, migration and invasion of breast cancer cells [69].

It has been demonstrated that natural-based molecules such as resveratrol can regulate the level of miRNAs. Resveratrol is a multi-role natural compound which has antitumor and anti-inflammatory roles. Besides this, it can also modulate miRNA expression. In BC cells, resveratrol regulates cancer cell growth by promoting the expression of miR-424-5p and inhibiting HNRNPA1. Moreover, HNRNPA1, which is related with tumor progression, is modulated by miR-424-5p [70].

For another subtype of BC, basal-like breast cancer [71], the expression of miR-424-5p is decreased in its tissues and cell lines, and the oncogene bicorticoid kinase 1 (DCLK1) is directly targeted to regulate tumor proliferation, invasion and migration. Patients with a low expression of miR-424-5p had higher clinical stages, larger tumor sizes and worse histological grades. In summary, miR-424-5p is a tumor suppressive miRNA in breast cancer that has therapeutic potential to enhance tumor immunity and inhibit the proliferation of tumor cells in BC.

### 3.6. Non-Small-Cell Lung Cancer

Non-small-cell lung cancer (NSCLC), as the most serious type of lung cancer with a high incidence, urgently needs new treatment strategies. Some studies [29] have shown that, in NSCLC tissues, miR-424-5p is poorly expressed, while lncRNA PVT1 and arginine methyltransferase 1 (CARM1) were highly increased. It is known that the combination of miRNAs and lncRNAs [72] can lead to the degradation of lncRNAs and affect the occurrence and development of tumors. Therefore, lncRNA PVT1 and CARM1 can bind to miR-424-5p. Overexpression of miR-424-5p inhibits the expression of CARM1 and increases the factors related to tumor progression and apoptosis, thus inhibiting tumor cell proliferation, migration and invasion and even improving the radiosensitivity of NSCLC. Similarly, YI [73] also showed that LINC00641 suppressed cancer in NSCLC by increasing the content of miR-424-5p. MiRNA and LncRNA are both key gene expression regulatory molecules [74], which play an important role in the occurrence and development of tumors. They can regulate each other with multiple sites and targets and interact and affect the performance of their respective functions. Therefore, in recent years, research on miR-424-5p has been increasingly related to lncRNAs. It is necessary to know that the disruption of the regulatory balance leads to tumorigenesis. Perhaps the key to treatment is to restore the balance of molecular regulation.

### 3.7. Osteosarcoma

Osteosarcoma is a malignant tumor that often occurs in adolescents and grows around the knee joint. It is the most common form of bone cancer [75]. Selvaraj [30] determined the function of melatonin in the osteosarcoma microenvironment. The experimental results confirmed that melatonin can inhibit tumor angiogenesis through the miR-424-5p/vascular endothelial growth factor A (VEGFA) axis, regulate the proliferation and migration of surrounding endothelial cells, the vascular morphology and the angiogenic growth factors and play a key role in tumor inhibition. Therefore, miR-424-5p can inhibit the proliferation, invasion and migration of cancer cells in osteosarcoma and provide new ideas for creating more effective treatments.

### 3.8. Nasopharyngeal Carcinoma

Nasopharyngeal carcinoma (NPC) is a malignant tumor originating from the mucosal epithelium of the nasopharynx. It is prevalent in southern China and Southeast Asia [76]. Finding molecular targets related to the malignant biological behavior of NPC is very important for improving the clinical treatment of NPC. Zhao collected 26 normal tissues and 40 NPC patient tissues [31]. It was found that the expression of miR-424-5p in NPC tissues was downregulated and negatively proportional to lymph node metastasis and clinical staging. It was also confirmed that AKT3 is the direct target of miR-424-5p and that miR-424-5p inhibits the proliferation, migration and invasion of NPC cells by reducing the expression of AKT3. The downregulation of miR-424-5p in NPC is closely related to the occurrence and development of NPC.

## 4. MiR-424-5p Acts as an Oncogene

MiR-424-5p can also improve cancer progression. In some specific cancers, miR-424-5p can bind to the target and promote tumor cell proliferation and migration. The mechanisms of miR-424-5p are briefly discussed.

### 4.1. Pancreatic Cancer

Pancreatic cancer is an invasive malignant tumor with high mortality. Because of its aggressive biological characteristics, the current treatment methods cannot greatly increase the prognosis of patients. Even surgical resection cannot significantly improve the survival rate of patients [77]. Therefore, it is necessary to explore unique molecular targets and biotherapy for pancreatic cancer. Wu et al. [32] showed that the expression of miR-424-5p was upregulated in pancreatic cancer. Dual-luciferase reporter gene detection showed that cytokine-induced signal transduction 6 (SOCS6) was the direct target of miR-424-5p. MiR-424-5p is significantly upregulated in pancreatic cancer and regulates the ERK1/2 signaling pathway through the negative regulation of SOCS6, which leads to the proliferation, migration and invasion of pancreatic cancer cells. Therefore, miR-424-5p may play a beneficial role in the diagnosis and treatment of pancreatic cancer.

### 4.2. Thyroid Cancer

Liu et al. [33] evaluated the expression of miR-424-5p in thyroid carcinoma and analyzed the miRNA dataset of The Cancer Genome Atlas (TCGA). The results showed that the expression of miR-424-5p in thyroid carcinoma was significantly higher than that in normal tissues. It was verified that the upregulation of miR-424-5p suppressed several downstream genes of the Hippo pathway in thyroid cancer cells. Dual-luciferase detection showed that WWC1, SAV1 and LAST2 were direct targets of miR-424-5p in thyroid cancer cells and were negatively correlated. Therefore, miR-424-5p suppresses the activity of Hippo signal transduction by targeting WWC1, SAV1 and LAST2, thus promoting lung metastasis and the anoikis resistance of thyroid cancer. Anoikis resistance refers to the capacity of tumor cells in suspension circumstances. Therefore, it represents the metastasis levels of cancer cells. The overexpression of miR-424-5p inhibits the activity of caspase-3 or -9 to promote anoikis resistance in thyroid cancer cells. This finding deepens our understanding of the molecular mechanisms underlying lung metastasis of thyroid cancer, which will provide new insights into the development of treatment strategies against lung metastases in thyroid cancer.

In addition, through bioinformatics analysis in the targeted regulatory network of miRNA–mRNA, it was shown that hsa-miR-424-5p may have the potential to synchronously regulate two central genes to reverse the process of papillary thyroid carcinoma. This may provide a new strategy for the treatment of thyroid cancer [78].

### 4.3. Gastric Cancer

Gastric cancer (GC) is one of the most common digestive tract malignant tumors. A large number of studies have shown that miRNAs play a key role in the development of GC by inhibiting the expression of target genes [79]. Studies by Jing [80] and others have shown that the overexpression of miR-424 promoted the proliferation and invasion of GC cells by targeting the LATS1 gene. Sequencing analysis showed that miR-424-5p was negatively and strongly correlated with the expression of LATS1, and therefore miR-424-5p could promote the proliferation of gastric cancer cells through LATS1. Similarly, Song et al. [34] examined the effect of miR-424-5p on the tumor growth of GC cells. The results showed that the overexpression of miR-424-5p promoted the proliferation of gastric cancer cells, while the knockdown of miR-424-5p could induce cell cycle arrest at the G0/G1 phase. Smad3 is known to be the direct target of miR-424-5p. MiR-424-5p promotes the proliferation of gastric cancer cells by targeting Smad3 through the TGF-β signaling pathway. Reducing the expression of miR-424-5p may be a new treatment that can be tried.

## 5. Controversial or Contradictory Research

Squamous cell carcinoma and colorectal cancer are cancers in which miR-424-5p can have both an inhibitory role and a stimulatory role. Through regulation of different pathways (CADM1, SOCS2, Smad7, OTX1), miR-424-5p can act to inhibit tumor progression.

### 5.1. Squamous Cell Carcinoma (SCC)

SCC [81] refers to a malignant tumor that occurs on the surface of the skin and the mucous membranes covered by squamous epithelium. The most common sites of SCCs are the skin, oral cavity, cervix, esophagus and vagina [82]. Because of its high invasiveness, the prognosis of SCC patients is still poor, and the mortality and morbidity of SCC are high [83]. Therefore, it is urgent to explore the progression and pathogenesis of squamous cell carcinoma. In recent years, among all types of cancer, most articles published on squamous cell carcinoma have explained the role of miR-424-5p. However, relatively contradictory results have been shown in regard to the mechanism and function of this miRNA.

Laryngeal squamous cell carcinoma (LSCC) accounts for more than 90% of laryngeal cancers and is a serious threat to the health of patients. Five years ago [84], to identify potential key genes, microarray data of mRNAs, miRNAs and lncRNAs from LSCC samples and adjacent normal samples were analyzed. It was found that hsa-miR-424-5p was differentially expressed in LSCC and targeted the regulation of multiple genes, which provided a foundation for future LSCC research at the molecular level. Li et al. [85] analyzed the expression of miR-424-5p in the tumors and surrounding normal marginal tissues of 106 patients with laryngeal LSCC. It was found that miR-424-5p was increased and that a higher level of miR-424-5p was positively correlated with advanced tumor stage, poor tissue differentiation and cervical lymph node metastasis. It was found that miR-424-5p suppressed the expression of CADM1, which is known to have an inhibitory effect on tumorigenesis [35]. Therefore, miR-424-5p downregulates the progression of LSCC by downregulating the level of CADM1 and promotes tumor proliferation, invasion and migration.

Similarly, oral squamous cell carcinoma (OSCC) is a common form of head and neck cancer that has the characteristics of local invasiveness, high recurrence and metastasis. Peng et al. [36] showed that miR-424-5p is upregulated in OSCC tissue and functions as a regulator of the IL-8/STAT5/SOCS2 feedback loop. IL-8 induces the expression of miR-424-5p by activating STAT5, while miR-424-5p targets and negatively regulates the expression of SOCS2. The downregulation of SOCS2 protein content results in a decrease in the cell proliferation rate and an increase in the activity of STAT5 [86], which promotes the tumorigenesis of OSCC. The role of the miR-424-5p oncogene in OSCC was also verified in another experiment. Exosomal CircGDI2 mediates the expression of tumor invasion inhibitor (SCAI) by sponging miR-424-5p. The decreased level of miR-424-5p can upregulate SCAI and reduce the proliferation, migration, invasion and glycolysis of OSCC cells [87].

Esophageal squamous cell carcinoma (ESCC) is one of the most common malignant tumors of the upper digestive tract and is also a more aggressive malignant tumor [88]. China has one of the highest incidences of esophageal cancer. To explore the mechanism of tumorigenesis and potential therapeutic targets of ESCC, studies [89,90] based on RNA sequencing and bioinformatics analysis have been performed to identify dysfunctional mRNAs, miRNAs and lncRNAs in ESCC. The noncoding RNA/protein coding interaction network was constructed, the signaling transduction pathway of dysfunctional genes was predicted, and hsa-miR-424-5p was shown to be the central miRNA in this network.

Ying [91] found that the expression of miR-424-5p is upregulated in ESCC tissues and that the overexpression of miR-424-5p can promote the proliferation and migration of ESCC through the targeted regulation of SIRT4. SIRT4 is a member of the SIRT family [37] and has been proven to play an important role in longevity and metabolism. Therefore, the conclusion in 2016 was that miR-424-5p exists as an oncogene in ECSS. However, four years later, new research showed the opposite trend to the above results. According to studies by Wang [92] and others, it was found that the expression of miR-424-5p decreased in ESCC tissues and cell lines. MiR-424-5p negatively regulates the expression of Smad7, while the Smad7 signal transduction pathway is known to promote the metastasis of advanced malignant tumors by participating in EMT. EMT is the first step of cancer invasion and metastasis [38]. In summary, miR-424-5p participates in the EMT of ESCC cells through the Smad7 signaling pathway, which leads to tumor proliferation, invasion and migration and exists as an antitumor factor in ESCC.

Cervical squamous cell carcinoma, which is mainly caused by distant metastasis, is a common clinical malignant tumor and the most common pathological type of cervical cancer [39]. One study proved that the expression of miR-424-5p was decreased in cervical squamous cell carcinoma and may target many cell cycle regulatory factors (CCND1, CCNE1, WEE1), indicating that miR-424-5p plays a biological role in cervical squamous cell carcinoma by regulating cell cycle factors and inhibiting cell proliferation and metastasis.

The conventional treatment for squamous cell carcinoma is mainly surgical resection in the early stage [93]. Radiotherapy and chemotherapy are the main treatments for late distant metastasis. To improve the cure rate of patients with squamous cell carcinoma, the study of targeted gene therapy has been a long-standing goal. Studies on the relationship between miR-424-5p and squamous cell carcinoma are emerging, and the experimental results may be slightly controversial. However, it is undeniable that miR-424-5p plays a vital role in the occurrence and development of squamous cell carcinoma, which should help in the preparation of a more promising treatment.

### 5.2. Colorectal Cancer

Colorectal cancer (CRC) originates from the colorectal mucosal epithelium and is one of the most common malignant tumors of the digestive system in the clinic. It has been found [94] that the increased expression of miR-424-5p in the serum of patients with CRC may be a potential noninvasive biomarker of CRC. Subsequently, some scholars [95] have predicted many targets in CRC tissue- and serum-specific miRNAs. According to the results of miRNA enrichment, miR-424-5p was related to more than ten colon cancer marker genes. According to recent research, the biological function of miR-424-5p in colorectal cancer is two-sided. On the one hand, Wei [96] performed a heatmap analysis of miRNA expression profiles of 619 CRC and 11 normal colorectal tissues through the TCGA database. It was found that the expression of miR-424-5p in CRC was upregulated and that overexpression of miR-424-5p could promote cell proliferation, while knocking down the expression of miR-424-5p could inhibit the proliferation of CRC cells. Similarly, through in vivo and in vitro experiments comparing adjacent normal tissue and cells, Dai [41] showed that miR-424-5p was significantly upregulated in CRC and promoted the proliferation and metastasis of CRC cells to act as an oncogene by directly targeting SCN4B. At the same time, it was also found that circulating exosomal miR-424-5p is a novel potential biomarker for the early diagnosis of CRC patients. The miR-424-5p target is significantly related to the regulation of sodium ion transport. The targeting effect of lncRNA on miR-424-5p has also been described [40]. The results showed that lncRNA FENDRR was poorly expressed in CRC tissues and cells, while low FENDRR inhibited cell proliferation and metastasis by targeting miR-424-5p. In summary, miR-424-5p plays the role of an oncogene in CRC. On the other hand, some studies have shown that the expression of miR-424-5p is downregulated in CRC, while its target gene human homeobox gene 1 (OTX1) is negatively correlated with miR-424-5p in colon cancer, and therefore the overexpression of miR-424-5p inhibits tumor proliferation and metastasis by downregulating OTX1. Another study also obtained similar results: the overexpression of miR-424-5p inhibited the viability and clone formation ability of colon cancer HT-29 cells and therefore inhibited the occurrence and development of tumor cells. The duality shown in CRC may be caused by different experimental conditions and environments. However, it is worth affirming that the disorder of miR-424-5p is closely related to the progression of colon cancer and provides potential methods for targeted gene therapy.

## 6. MiR-424-5p in the TME

The tumor microenvironment (TME) is a systematic concept. Tumor cells recruit fibroblasts, vascular endothelial cells, lymphocytes, macrophages, neutrophils and other immune cells to form the tumor matrix. These nontumor cells interact with the extracellular matrix and various cellular and extracellular components, inducing the TME into an immunosuppressive state [97]. For example, programmed cell death ligand-1 (PD-L1), which is overexpressed on cancer cells, is a key mediator of the immunosuppressive microenvironment, interacting with the PD-1 receptor in cancer, which plays an important role in tumor immune escape. PD-L1 is a specific type I transmembrane protein expressed on tumor cells, natural killer cells (NKs), T cells and dendritic cells (DCs) [98]. Researchers have shown that upregulation of miR-424-5p can promote an inflammatory response by increasing the secretion of inflammatory cytokines such as IFN-γ, TNF-α and IL-6 and reducing the production of the strong immunosuppressive cytokine IL-10 [99]. MiR-424-5p can be effectively transferred and have an effect through extracellular vesicles (EVs). The EV-mediated delivery of miR-424-5p inhibits PD-L1 signal transduction, creating an inflammatory microenvironment in triple-negative breast cancer (TNBC), which increases the secretion of proinflammatory cytokines and reduces the secretion of anti-inflammatory cytokines. Therefore, the inflammatory microenvironment enhances tumor cell apoptosis in vitro and inhibits tumor growth in vivo. This finding may provide a new way to block the immune checkpoint by EVs, enhance the immune response to tumor cells and provide more new ideas and strategies in tumor immunotherapy [100].

## 7. MiR-424-5p and Chemotherapy

With the discovery of miRNA and its important regulatory role in many diseases, it is very important to explore and understand the possibility of miRNA as a therapeutic drug. In recent years, studies on animals from different laboratories have provided data to enhance the applicability of miRNA in clinical cases. Thus, miRNA can be considered an innovative treatment known as miRNA therapy [101]. However, this treatment is still in its early stages of exploration. Although a small number of miRNA therapies have recently been put into clinical trials, this technology is still not sufficiently mature. Chemotherapy and radiotherapy are still the main treatments for tumors. Moreover, miRNAs also play an important role in the regulation of radiotherapy and chemotherapy. How miRNA works in synergy with other recognized chemotherapeutic drugs can be proven to demonstrate that miR-424-5p can be an anticancer agent in some tumors. Additionally, studies have confirmed that miR-424-5p can improve the sensitivity to chemotherapeutic drugs [16,102].

Paclitaxel is an organic compound extracted and isolated from natural yew plants and is one of the most widely used anticancer drugs in the world. MiR-424-5p induces the sensitivity of MDA-MB-231 breast cancer cells to paclitaxel and inhibits cancer cell proliferation by inducing apoptosis. The specific mechanism [103] may be that miR-424-5p enhances the chemosensitivity of MDA-MB-231 breast cancer cells by regulating the PD-L1 and PTEN/mTOR axes and apoptosis-related factors, including p53, caspase 3, Bcl-2, and Bax, and the proliferation-related gene c-Myc. Previously, paclitaxel was often used in combination with another chemotherapy drug to improve the effectiveness of treatment, while the combination of miR-424-5p and paclitaxel represented a new anti-BC strategy.

Similarly, 6-thioguanine (6-TG) has been used as an anticancer agent and immunosuppressant in medical practice for many years, and it has been proven that 6-TG can inhibit the growth of MCF-7 breast cancer cells. To explore the mechanism of action, the ceRNA network was constructed by comparing, predicting and integrating the differentially expressed RNAs. MiRNA is the core factor of the ceRNA network and an important tumor marker. In this study, miR-424-5p was identified as the core regulatory element in the ceRNA network. The qPCR results showed that after 6-TG treatment of MCF-7 cells, the expression level of miR-424-5p was decreased and regulated specifically by GADD45G and CCND3 to change the apoptosis and cell cycle of the breast cancer cells. The effect of 6-TG therapy on MCF-7 breast cancer cells establishes a theoretical foundation for the further study of the molecular mechanism of breast cancer regulation and the screening of potential therapeutic targets [104].

There is another chemotherapeutic drug, ginsenoside Rg3, which is the active component of Panax ginseng. Ginsenoside Rg3 [105] is a potential molecular target and therapeutic indicator in metastatic breast cancer. Ginsenoside Rg3 acts negatively in cancer cells by regulating a variety of noncoding RNAs. Rg3 induces hypermethylation and downregulation of lncRNA ATXN8OS, while lncRNA ATXN8OS mainly stimulates key oncogenes through sponging miR-424-5p, thus inhibiting cancer cell proliferation [106].

The chemotherapy mentioned above only involves breast cancer, and the conjecture that miR-424-5p can improve chemosensitivity needs to be verified in many other tumors. In addition, the effects of miR-424-5p need to be further proven in animal models in order to reach the goal of initiating clinical trials.

## 8. Conclusions

Researchers have been looking for new ways to treat cancer. With the progress in scientific research and technology, miRNAs have been shown to have a variety of gene expression functions in controlling cancer and other important diseases, making miRNAs ideal candidates for therapeutic applications. This article reviews the latest and most powerful research on miR-424-5p in tumors, which is helpful for finding new ways to treat cancer.

MiR-424-5p can be detected in human body fluids. Compared with the body fluids of healthy people, the concentration of miR-424-5p in body fluids of cancer patients is usually different and is abnormally expressed in cancer. Cancer cells can also secrete miR-424-5p into the extracellular matrix through EVs, which affects other cells in the TME.

The expression level of miR-424-5p is related to a variety of important tumor clinical parameters, including tumor size, multiple nodules, vascular invasion, TNM stage and overall survival, and the expression is upregulated or downregulated depending on the type of tumor. The function of miR-424-5p is very complex and contradictory. On the one hand, miR-424-5p may negatively regulate tumor cell proliferation and other malignant biological behaviors in tumors. On the other hand, miR-424-5p can also promote malignant changes. However, miR-424-5p has shown an inhibitory effect on tumor proliferation in most cases.

Many studies not only provide basic theoretical support for the regulatory role of miR-424-5p in tumors but also provide new schemes and ideas for clinical treatment at the molecular biology level. However, tumors are heterogeneous, and the role of miR-424-5p is different in different tumors, which also adds a serious challenge to research into miR-424-5p. Therefore, the biological effects of miR-424-5p in tumor progression need to be further studied. However, it is undeniable that miR-424-5p has the potential to be used as an anticancer agent in some tumors and plays an important role in the fight against cancer. We believe that, with the continuous research and development of miRNAs in the near future, these miRNAs can be used in clinical treatments.

There are still some kinds of cancers that have not been well studied and the relationship between cancers and miRNAs has not been explored. In hematological cancer, miRNA-424-5p has been proved to enhance HOXA expression by bioinformatics analysis, which can be oncogenic and promote the cancer cell proliferation and migration in acute myeloid leukemia. However, the specific mechanisms of how miRNAs regulate in hematological cancer are not known. Therefore, this can be a new direction for scientists to research.

Although many studies have been made into miRNAs, there is still a long way for scientists to apply miRNAs to clinical trials. It is expected that researchers and scholars will explore the role of miR-424-5p in vivo because most articles about miR-424-5p are focused on research in vitro. If miR-424-5p is expected to treat cancers in clinic, it is necessary to figure out how it works in the human body. Based on the mechanisms of miR-424-5p, new drugs that work on new targets can be made.

## Figures and Tables

**Figure 1 ijms-23-04037-f001:**
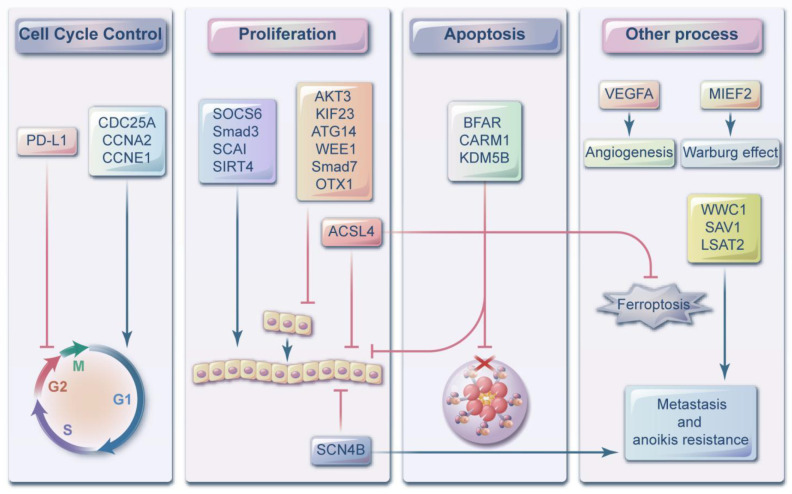
The ways that miR-424-5p regulates cancer cell progression can be categorized into 4 mechanisms. (i) By controlling cell cycle. (ii) By promoting or inhibiting cancer cell proliferation, (iii) By regulating tumor cell apoptosis (iv) By acting on the processes that cancer cells have, such as the Warburg effect, metastasis, anoikis resistance, ferroptosis and angiogenesis. The targets of miR-424-5p can not only participate in one ways, but can have different roles in different mechanisms.

**Figure 2 ijms-23-04037-f002:**
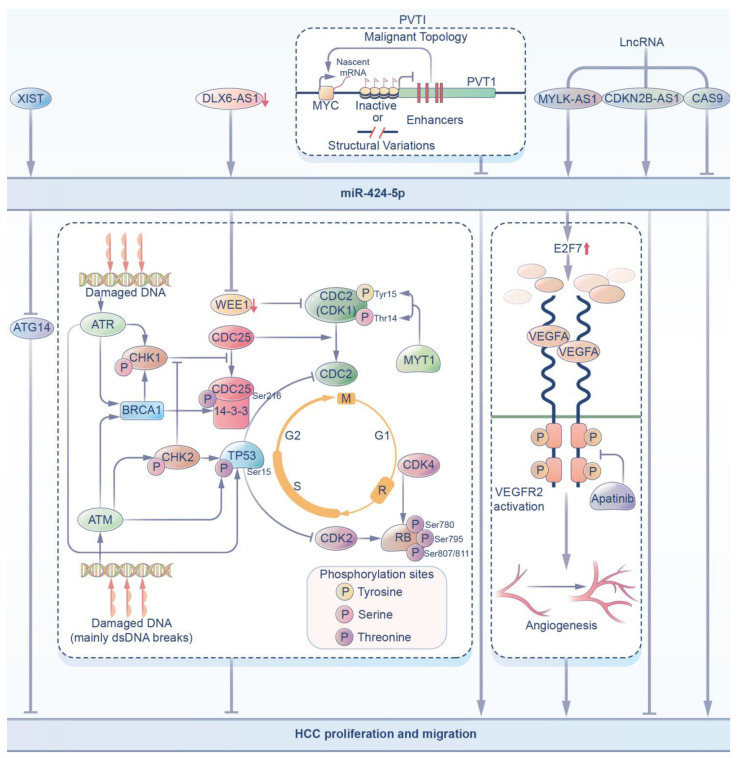
In HCC, miR-424-5p can be regulated by upstream pathways and regulate downstream pathways. (i) XIST can increase the expression of miR-424-5p, which inhibits ATG14, inhibiting HCC proliferation and migration. (ii) The decrease in DLX6-AS1 can target miR-424-5p to inhibit the expression of WEE1, which inhibits HCC proliferation and migration. (iii) PVT1 inhibits the expression of miR-424-5p by regulating the p53 signaling pathway to promote HCC proliferation. (iv) LncRNAs, including MYLK-AS1, CDKN2B-AS1, and CAS9, can affect the expression of miR-424-5p to regulate HCC progression.

**Figure 3 ijms-23-04037-f003:**
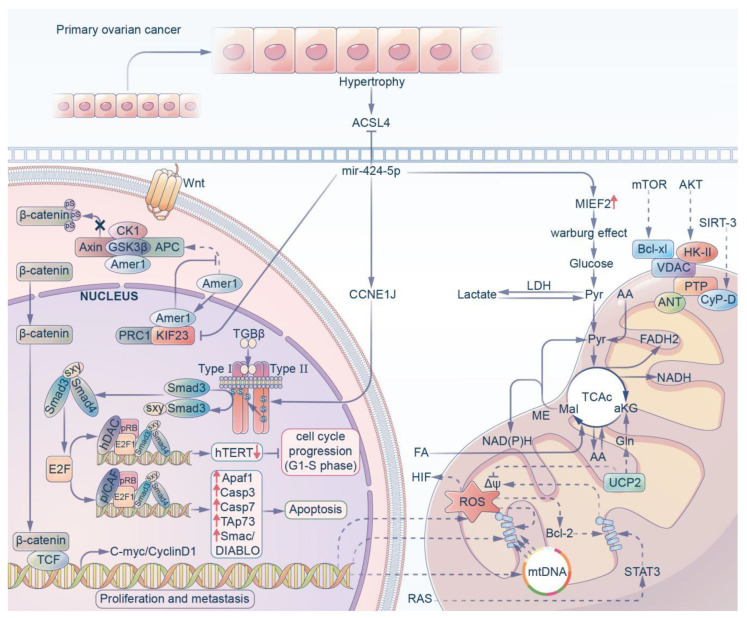
MiR-424-5p can regulate ovarian cancer progression in 4 different ways. (i) MiR-424-5p can inhibit ACSL4 to prevent hypertrophy of primary ovarian cancer. (ii) MiR-424-5p decreases the expression of KIF23 to inhibit the proliferation and migration of ovarian cancer. (iii) MiR-424-5p targets CCNE1J directly to activate the E2F1-pRb signaling pathway, which results in an antitumor effect. (iv) MiR-424-5p increases the expression of MIEF2 to facilitate the Warburg effect.

**Table 1 ijms-23-04037-t001:** Role of miR-424-5p in various cancers.

Type of Cancer	Effect of mir-424-5p	Mechanisms	Reference
Hepatocellular carcinoma (HCC)	Tumor suppressor	Inhibit ATG14, WEE1, activate E2F7	[21,22]
Intrahepatic cholangiocarcinoma (ICC)	Tumor suppressor	Regulate ARK5, inhibit mTOR phosphorylation	[23]
Ovarian cancer	Tumor suppressor	Inhibit ACSL4, KIF23, activate CCNE1J, MIEF2	[24,25]
Cervical cancer	Tumor suppressor	Activate KDM5B-Notch	[26]
Malignant tumors of the nervous system	Tumor suppressor	Target BFAR and ALK receptor directly, increase βFGF	[27]
Breast cancer	Tumor suppressor	Target PD-L1, regulate PTEN/PI3K/AKT/mTOR	[28]
Non-small cell carcinoma (NSCLC)	Tumor suppressor	Inhibit CARM1	[29]
Osteosarcoma	Tumor suppressor	Target VEGFA, inhibit angiogenesis	[30]
Nasopharyngeal carcinoma	Tumor suppressor	Inhibit AKT3	[31]
Pancreatic cancer	Oncogene	Inhibit SOC6 to regulate ERK1/2	[32]
Thyroid cancer	Oncogene	Target WWC1, SAV1, LSAT2 to inhibit Hippo	[33]
Gastric (stomach cancer)	Oncogene	Target Smad3 to regulate TGF-β	[34]
Laryngeal squamous-cell Carcinoma	Tumor suppressor	Inhibit CADM1	[35]
Oral squamous-cell Carcinoma	Tumor suppressor	Activate IL-8 to inhibit SOCS2; activate SCAI	[36]
Esophageal squamous-cell Carcinoma	Evidence for both	Oncogene: target SIRT4;Tumor suppressor: inhibit Smad7	[37,38]
Cervical squamous-cell carcinoma	oncogene	Target CCND1, CCNE1, WEE1 to regulate cell cycle	[39]
Colorectal cancer	Evidence for both	Oncogene: target SCN4B;Tumor suppressor: inhibit OTX1	[40,41]

## Data Availability

All data are available in the manuscript.

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
