# Peer review of "Sequence Requirements for miR-424-5p Regulating and Function in Cancers"

_ijms, 2022, doi:10.3390/ijms23074037_

Round 1
Reviewer 1 Report
I thank to the editors for the opportunity to review this work, beside I would also like to congratulate the authors for the made effort. This review aims to summarize the roles of miR-424-5p in different kinds of cancers, making a mechanistic analysis of the role of miR-424-5p in various tissues through target gene verification and pathway analysis. In addition, the novelty of this review is that to date there are no reviews summarizing all the findings related to miR-424-5p and its involvement in cancer. Nevertheless, and in spite of the significant amount of work performed, some important issues have to be consider.
General Comments:
- In general, the review is well written and reads easily.
- In “Introduction” section, the authors say that “miRNAs have also been observed to play roles as oncogenes or tumor suppressor genes to participate in tumor cell proliferation, differentiation, apoptosis and other biological processes”. I totally agree with this assertion, so I suggest to them including a first general figure showing all the intracellular pathways in which miR.424-5p is involved and which it will trigger as a cancer-associated mechanism including up- and down-regulated expression. I consider that this review provides a lot of information and it would be enlightening to summarize it in a general figure.
- In this regard, I did a quick search in Pubmed and I found some studies focusing on the role of miR.424-5p in cell death not only in apoptosis, as mentioned in this review, but in other types such as ferroptosis and anoikis in ovarian and endometrial cancers. In my opinion, these functions should be included in the review, at least named as alternative mechanisms of action.
- Authors say that “Upregulated LINC00511 stimulates the proliferation of ovarian cancer cells by inhibiting apoptosis, while this long noncoding RNA inhibits the expression of a series of miRNAs and increases the expression of on-cogenes, including miR-424-5p. Therefore, miR-424-5p shows an active anticancer effect in OV”. But I think that this information is contradictory, because of this, miR-424-5p should show an active tumoral effect in OV, not anticancer effect.
- Authors say that “studies have confirmed that miR-424-5p can improve the sensitivity to chemotherapeutic drugs”, but no references are provided of these studies. They should be added.
Minor comments:
P3: Authors should define the meaning of abbreviation “lncRNAs”.
P4,5,6: Authors should correct the bold font of “in vivo” and “in vitro”.
P5: Authors should correct the word “Topolpgy” by “Topology” in Figure 1.

Author Response
Response to Reviewer 1 Comments
The response to review 1 comments are highlighted with yellow color(Please see the attachment).
Point 1: In“Introduction” section, the authors say that “miRNAs have also been observed to play roles as oncogenes or tumor suppressor genes to participate in tumor cell proliferation, differentiation, apoptosis and other biological processes”. I totally agree with this assertion, so I suggest to them including a first general figure showing all the intracellular pathways in which miR.424-5p is involved and which it will trigger as a cancer-associated mechanism including up- and down-regulated expression. I consider that this review provides a lot of information and it would be enlightening to summarize it in a general figure.
Response 1: Because I have concluded a table in the introduction part and it is difficult to describe the whole mechanisms about the regulation of miR-424-5p in one figure, I provided a rough figure to conclude the basic ways about how miR-424-5p regulates and the roles their targets play in tumor cells.
Point 2: In this regard, I did a quick search in Pubmed and I found some studies focusing on the role of miR.424-5p in cell death not only in apoptosis, as mentioned in this review, but in other types such as ferroptosis and anoikis in ovarian and endometrial cancers. In my opinion, these functions should be included in the review, at least named as alternative mechanisms of action.
Response 2: I searched the article about the miR-424-5p and its role in ferroptosis and anoikis and I found that these articles were already included in my manuscript, they are [24] and [33], respectively. But at that time I did not describe the mechanisms of ferroptosis and anoikis with detail. Therefore, I make a supplement description in the revised version.
Point 3: Authors say that “Upregulated LINC00511 stimulates the proliferation of ovarian cancer cells by inhibiting apoptosis, while this long non-coding RNA inhibits the expression of a series of miRNAs and increases the expression of on-cogenes, including miR-424-5p. Therefore, miR-424-5p shows an active anticancer effect in OV”. But I think that this information is contradictory, because of this, miR-424-5p should show an active tumoral effect in OV, not anticancer effect.
Response 3: I think it is a comprehensive problem. The long non-coding RNA inhibits the expression of a series of miRNAs, including miR-424-5p, and increases the expression of on-cogenes. The miR-424-5p is inhibited and the on-cogenes expression is increased, so the miR-424-5p shows an anticancer effect.
Point 4: Authors say that “studies have confirmed that miR-424-5p can improve the sensitivity to chemotherapeutic drugs”, but no references are provided of these studies. They should be added.
Response 4: I have added the references in the revised manuscript.
Point 5: Minor comments:
P3: Authors should define the meaning of abbreviation “lncRNAs”.
P4,5,6: Authors should correct the bold font of “in vivo” and “in vitro”.
P5: Authors should correct the word “Topolpgy” by “Topology” in Figure 1.
Response 5: I have changed these details in the revised manuscript.

Reviewer 2 Report
Authors present a manuscript addressing the sequence requirements for miR‐424‐5p regulating and function in cancers. The aim of this review was to discuss how miR-424-5p plays a role in different kinds of cancers from different stages of tumors, including its roles in (i) promoting or inhibiting tumorigenesis, (ii) reguating tumor development in the tumor microenvironment and (iii) participating in cancer chemotherapy. This review provides a deep discussion of the latest findings on miR-424-5p and its importance in cancer, as well as a mechanistic analysis of the role of miR-424-5p in various tissues through target gene verification and pathway analysis. This is a very well-written manuscript but I have reviewed the manuscript and have the following comments/suggestions:
- A section of future perspectives and a little bit more discussion would be appreciated.
- I suggest to describe deregulation of miR-424-5p in hematological cancers.
- I recommend to add following articles:
- a) Xie D, Song H, Wu T, Li D, Hua K, Xu H, Zhao B, Wu C, Hu J, Ji C, Deng Y, Fang L. MicroRNA‑424 serves an anti‑oncogenic role by targeting cyclin‑dependent kinase 1 in breast cancer cells. Oncol Rep. 2018 Dec;40(6):3416-3426.
- b) Wu J, Yang B, Zhang Y, Feng X, He B, Xie H, Zhou L, Wu J, Zheng S. miR-424-5p represses the metastasis and invasion of intrahepatic cholangiocarcinoma by targeting ARK5. Int J Biol Sci. 2019 Jun 4;15(8):1591-1599.
- c) Dastmalchi N, Baradaran B, Banan Khojasteh SM, Hosseinpourfeizi M, Safaralizadeh R. miR-424: A novel potential therapeutic target and prognostic factor in malignancies. Cell Biol Int. 2021 Apr;45(4):720-730.
Author Response
Response to Reviewer 2 Comments
The response to review 1 comments are highlighted with green color(please see the attachment).
Point 1: A section of future perspectives and a little bit more discussion would be appreciated.
Response 1: I have added some future perspectives and some directions to explore for the researchers in the discussion section.
Point 2: I suggest to describe deregulation of miR-424-5p in hematological cancers.
Response 2: I searched the article about the miR-424-5p and its role in hematological cancers, but the results are not expected. The hematological cancers include leukemia, lymphoma and myeloma. The results about miR-424-5p and leukemia are 2 bioinformatic analysis and is not specific about miR-424-5p. There is no result about miR-424-5p and myeloma. The result about miR-424-5p and lymphoma is only one. Because the results about the regulation of miR-424-5p in hematological cancers are so few, I roughly concluded the results in discussion part and I think it can be as a future direction for scientists to explore. Therefore, I concluded it in the discussion part.
Point 3: I recommend to add following articles:
- Xie D, Song H, Wu T, Li D, Hua K, Xu H, Zhao B, Wu C, Hu J, Ji C, Deng Y, Fang L. MicroRNA424 serves an antioncogenic role by targeting cyclindependent kinase 1 in breast cancer cells. Oncol Rep. 2018 Dec;40(6):3416-3426.
- b) Wu J, Yang B, Zhang Y, Feng X, He B, Xie H, Zhou L, Wu J, Zheng S. miR-424-5p represses the metastasis and invasion of intrahepatic cholangiocarcinoma by targeting ARK5. Int J Biol Sci. 2019 Jun 4;15(8):1591-1599.
- c) Dastmalchi N, Baradaran B, Banan Khojasteh SM, Hosseinpourfeizi M, Safaralizadeh R. miR-424: A novel potential therapeutic target and prognostic factor in malignancies. Cell Biol Int. 2021 Apr;45(4):720-730.
Response 3: I have added the articles a) and c) in the revised manuscript, they are [68] and [70], respectively. And the b) article was already included in the previous manuscript [23].
